# Soccer and Disability, Is It Possible? Analysis of the Learning and Coaching Context in Spain

**DOI:** 10.3390/sports11090161

**Published:** 2023-08-28

**Authors:** Antonio Burgos-García

**Affiliations:** Department of Didactic and School Organization, Faculty of Education, University of Granada, 18012 Granada, Spain; aburgos@ugr.es

**Keywords:** disability, professional identity, coach education profile, grassroots-soccer coach, soccer

## Abstract

Coaching a youth soccer player is important, and the coach’s role is key. Actually, there is no profile or coaching program for grassroots-soccer coaches that favor the practice of soccer and disability, according to different research and experts. The main purpose of this study is to identify and analyze the professional profile of the grassroots-soccer coach who has soccer players with disabilities (learning and coaching context). This research applies a quantitative method, specifically, non-experimental, cross-sectional, descriptive, and inferential methodology. The sample of analysis is the staff members of the professional soccer clubs of LaLigaSantander Genuine (Spain). An important result is that half of the grassroots-soccer coaches have not received specific education to coach youth soccer players with disabilities. Finally, one important conclusion of this research is that by generating a climate of trust and empathy, grassroots-soccer coaches improve the performance of their soccer players with disabilities by recognizing and understanding their emotional states.

## 1. Introduction

The sports coaching process of all youth soccer players has been the subject of discussion, especially in the context of grassroots-soccer, where the coach’s role emerges as one of the key factors in the coaching process [1]. The need to improve the profile and coaching of coaches for sports teaching has led to several studies and research to analyze the coaching process [2,3,4].

The coach education process is key, as it is the coach’s responsibility to provide competent guidance to their group concerning different sports techniques and effective strategies and behaviors, such as creativity, autonomy, independence, and disability awareness [5]. Regarding this situation, a problem is that the coach feels unprepared to coach soccer players with disabilities because the coach has to understand the sport and their soccer players [6].

Also, this coaching process would generate socio-cognitive situations (learning of values—support and social commitment, respect, etc.—acquisition of skills, healthy behaviors, etc.) through which soccer players can discover the positive benefits of their sport participation and, in parallel, key factors to enable the employability [7].

In this sense, it should be ensured that the learning context of the coaching process of the coaches could respond to different profiles, taking as a reference the contribution made by Fraile Aranda et al. [8] and Petrovska et al. [9]. Firstly, within the coaching of coaches is the profile “dialoger” (D). This profile includes knowing how to communicate to achieve a good environment and climate in coaching and having the good interpersonal relations of being assertive, respectful, and empathetic, with an ability to resolve conflicts.

Secondly, a “critical” (C) profile capable of generating a climate of reflection and analysis of the coaching, enabling a critical attitude and analysis of the actions and behavior of the soccer players. This profile demonstrates self-criticism from an assessment perspective concerning the design and development of coaching programs.

Thirdly is an innovative profile (I), defined as “comprehensive”. This profile maintains a creative perspective for the learning of individual and collective technical and tactical behaviors from an active methodology. The fourth profile would be related to the idea of a “collaborator” (CLL). This profile highlights the need to be able to organize based on a division of labor in which there is a distribution and sharing of tasks and roles. Finally, there is the traditional profile (T) (application of a mastering or “direct command” coaching style) and the technological profile (TCH) (the priority is to seek efficiency in their practice, using technological means and scientific knowledge as a reference in their coaching style).

In grassroots-soccer, the skills and abilities expressed in each profile are vital for the coach. These skills and abilities allow for an adequate climate of understanding, trust, and security between the group and the object of the coaching. Grané [10] points out that this situation would mean a very significant performance at the technical, sporting, and emotional levels of the group, especially when there are participants with some degree of disability. Coaching these skills and abilities in a natural way favors a key educational and experiential basis in the development and relations with the socio-community environment.

Camacho et al. [7] specify that it is necessary to stimulate individual factors (expressions of positive feedback behavior, instruction, democratic behavior, confidence, satisfaction, etc.) and contextual factors (resources, funding, spaces, etc.) generated by these skills and abilities because they have an impact on the personal development of the youth soccer players when among its members there are soccer players with disabilities, as these factors will increase the probability of achieving, among other aspects, quality employment.

The learning and coaching context in grassroots-soccer is the space where individual and contextual factors can be stimulated for soccer players, especially those with a certain degree of disability [11]. These authors conceive these factors as a tool for improvement that must be systematic, planned, organized, and structured beforehand by the coach to improve safety, order, variety, efficiency, improvement, and coordination of the work, allowing soccer players to acquire a progressive learning process, being consistent with the coaching process [12].

Within this coaching process, the idea is expressed that it is also an ideal environment for educating in values (tolerance, respect, responsibility, empathy, equality, etc.), as it is a playful activity that generates opportunities due to the number of skills, abilities, and conflicts that must be resolved once the coaching action has begun [13].

The type and quality of the values acquired will depend on the agents of socialization in the sporting environment in which they develop, with the coach having the greatest impact, playing a key role in the development and coaching of individuals and athletes [1,14,15].

In this sense, the coach’s role is defined as a model or reference that exercises great leadership and goes beyond the sporting context [16]. It is interesting to know the management style, the type of social support and reinforcement, the method of teaching, and the contents provided, as the coach will be decisive in the overall performance of the soccer players, especially those with some degree of disability.

In generating success and quality in the coaching and experiential processes of the soccer players, the coach must have as a reference the planning, development, and control of coaching, as well as contributing to the development of “soft skills” [9,17].

Also, the curriculum for coaches should promote a competence profile, as well as a professional profile. This curriculum should establish common goals, processes, and methods to be carried out that generate roles and benefits for each of the members of the group, in addition to putting strategies to emotionally activate these people into practice so that they comply with the set goals, eliminating possible problems [18] and, above all, emphasizing the following four areas to be developed [19]:▪Cognitive and learning: Selective attention; Sustained attention; Error detection; Learning potential;▪Functional: Ability to systematize tasks; Planning and organization; Resistance to repetitive tasks; Problem solving; Ability to ask for help when needed; Manipulative skills;▪Communicative: Comprehension of verbal instructions; Comprehension of written instructions; Oral expression; Written expression;▪Social and attitudinal: Relationship with professionals; Ability to work in a team; Respect and care for materials/tools; Level of responsibility; Level of initiative; Acceptance of criticism; Level of autonomy; Acceptance of authority; Level of flexibility and adaptation to changes; Level of assertiveness and capacity for empathy.

In these areas, it is important to highlight the positive influence that sport can have on the coaches [20], especially at the level of assertiveness and empathy capacity because this means that the coach increases their awareness of the most effective coaching strategies for their soccer players [21], generating a high level of commitment to learning, performance, and a higher degree of well-being [22]; especially, establishing a positive sports climate where the coach possesses a higher level of morality, communicating more successfully and encouraging the forging of empathetic and assertive motivational relationships with their players [23].

Different studies have shown that, currently, there are no professional coaching programs or specific coaching profiles for coaches that favor the practice of soccer or sport in general with soccer players with disability [24]. For this reason, different questions arise in this research. In this sense, what type of coach coaches our young players? Another important question is, what values and attitudes should coaches who coach young players with disabilities have? With this last question in mind, are cognitive empathy and assertiveness key values and attitudes for improving the performance of young players with disabilities? And if so, is it important to include these values and attitudes in the education of a grassroots-soccer coach? The answers to these questions can be generated from the main objective of our study. Therefore, from our research, it is necessary to identify and analyze the coaching profile of LaLigaSantander Genuine grassroots-soccer coaches (learning and coaching context) who coach soccer players with disabilities in Spain. In this way, the following specific aims are established:▪Analyzing the type of grassroots-soccer coach that coaches in the professional soccer clubs league called LaLiga Santander Genuine (Spain);▪Knowing the type of values and attitudes of a grassroots-soccer coach with disabled soccer players;▪Identifying the level of cognitive empathy and assertiveness according to the profile of the grassroots-soccer coach who coaches with clubs with soccer players with some disability.

## 2. Materials and Methods

### 2.1. Design

This research applies a quantitative, non-experimental, cross-sectional, descriptive, and inferential methodology. This study analyses the data of the variables under study without establishing any manipulation (natural context) [25]. In this way, it seeks to “specify the properties, characteristics, and profiles of people, groups, communities, processes, objects, or any other phenomenon subject to analysis” [25] (p. 92).

### 2.2. Sample

The sample of this study is composed of coaching staff members of the professional soccer clubs of LaLigaSantander Genuine (Spain) (36 clubs in the 21/22 season, with a minimum of three coaches per club). Finally, in the study, the population was set at 99 coaches, but 40 coaches participated (31.7%) with a mean age of 35.95 years (S.D. = 9.46, range = 22–57), of which 25% were women (10) and 75% men (30). In this sense, it is important to highlight that only 4 coaches (10%) work with groups that have a player with a disability.

### 2.3. Variables and Instruments

Three key aspects were analyzed regarding the coaching process of grassroots-soccer coaches, using as a dependent variable the profile of the adapted soccer coach found in grassroots coaching. Furthermore, this study analyzed the level of empathy and assertiveness of the coaches who work with disabled soccer players and, as independent variables, the socio-demographic variables that describe the coaching reality.

A total of three instruments were used. Firstly, the questionnaire concerning coaching models was used to determine the coach’s coaching profile [26]. The purpose of this instrument is to analyze the dimensions that make up the different coach profiles. The questionnaire is composed of 46 items. In this instrument, the coach’s orientation is identified in six factors: traditional, technological, innovative, collaborative, dialogic, and critical [12]. For each factor, Cronbach’s Alpha coefficient presented a reliability index of 0.818, 0.829, 0.622, 0.939, 0.711, and 0.787, respectively. The questionnaire uses a five-point Likert-type response format (1 = Not important and 5 = Very important).

Concerning identifying the value system regarding the coach’s attitudes towards interactions with the soccer players, the instrument used was self-reports of attitudes and values in social interactions (ADCAs) where “self-assertiveness” (AA) (degree or level of respect and consideration towards one’s own feelings, ideas, and behaviors) and “hetero-assertiveness” (HA) (degree or level of respect and consideration towards the feelings, ideas, and behaviors of others) make possible to establish the profile of attitudes and values of coaches through the type of assertive style in the coaching process in grassroots-soccer. Each instrument has a four-point Likert-type response format (0 = always or almost always and 3 = never or seldom). The reliability coefficients for this sample were α = 0.807 for the AA and α = 0.798 for the HA.

Finally, the cognitive and affective empathy test (TECA) is used to analyze the empathy of the grassroots-soccer coach, providing information on both the cognitive components of empathy (perspective-taking and emotional understanding) and the affective components (empathic stress and empathic joy) [27]. The reliability coefficients for the sample in each of them were 0.786, 0.673, 0.805, and 0.619, respectively. A five-point Likert-type agreement scale (1 = Strongly Disagree and 5 = Strongly Agree) was used as the response format.

### 2.4. Procedure

To analyze the profile of LaLigaSantander Genuine coaches working with young soccer players with disabilities, the process consisted of several phases. Firstly, once the study was authorized by the Human Research Ethics Committee of the University of Granada (Spain), code 2368/CEIH/2021, the research team emailed all soccer clubs requesting their collaboration. The email informed the soccer clubs of the purpose of the study and also guaranteed the anonymity and confidentiality of the data. Likewise, in this email, the coaches were informed of the characteristics of the study by signing an informed consent form before the start of data collection. Then, they were provided with the different questionnaires through the GoogleForms tool, in which, once answered, the data were exported to an Excel sheet. Subsequently, the data collected were imported into a single database in Excel format. Finally, the final data analysis used the SPSS statistical program (IBM Corp. Released 2021. IBM SPSS Statistics for Windows, Version 28.0., IBM Corp.: Armonk, NY, USA).

### 2.5. Data Analysis

In this study, a descriptive analysis was carried out using the mean, median, and standard deviation extracted from the coach profile. As the sample that participated in this research was less than 50, it was possible to contrast the normality with the Shapiro-Wilk test [28]. For homoscedasticity (homogeneity of variances), that is, to verify that the data come from a normal distribution and see that the variances of the groups of coaches are equal (homogeneity of variance contrasts), we used the Levene test as they allowed us to identify the relationship between the variables and the differences between groups [29]. According to these authors, a bivariate Pearson correlation analysis was performed to determine the presence of linear relationships between coach profiles, empathy, and assertiveness. In addition, a grouping of the subjects in the factors was made, which allowed ANOVA analyses to be performed for the different profiles. For assertiveness, groups were balanced at 33.3%, while for empathy they were established according to the classification provided by the questionnaire. The low and extremely low groups were unified due to the scarcity of participants assigned to them, even having to be eliminated for the post hoc contrasts since, in some cases, there was only one subject. Bonferroni statistic was used for the contrasts because the number of comparisons was small [30,31,32]. The significance level concerning the number of statistical tests performed simultaneously on the data set was *p* < 0.05.

## 3. Results

In the following, we present the research results regarding the general purpose and, therefore, the aims of this study.

### 3.1. Descriptive Analysis of the Grassroots-Soccer Coach

Descriptive results show that 47.5% (19) of the coaches were part of a full-time LaLigaSantander Genuine club, 41.3% (16) coached in parallel with another club in the same competition, and 11.2% (5) carried out their professional work with more than 2 clubs. The sample surveyed expressed an average professional coaching experience as grassroots-soccer coaches of 12.75 years (S.D. = 9.242 and range = 0–30 years) but in the exercise of this profession with disabled soccer players is 6.13 years (S.D. = 5.431 and range = 0–22 years).

Grassroots-soccer coaches have a broad educational background. The majority of them, 50%, come from the university environment (from the Education Degree—*n* = 7—and Sciences of Physical Activity and Sport—*n* = 6—as well as Psychology—*n* = 3) followed by Vocational Training, specifically, Technician in Physical Sports Activities Degree (27.5%). The other coaches have only Compulsory Education (5%) and High School (5%). In any case, the key to this descriptive analysis is given by those who have or have not received specific coaching to develop their professional work with soccer players with disabilities. In this sense, 57.5% (23) stated that they had received this coaching, which was not formal, as it was provided by national federations, foundations, or associations. The rest of the coaches, specifically 42.5% (17), stated that they had not received any coaching in this sense.

### 3.2. Knowledge of the Type of Values, Attitudes, and Profiles of a Grassroots-Soccer Coach with Disabled Soccer Players

Regarding the values and attitudes of the different coaches surveyed, it has been observed that the coaches develop a direct interpersonal behavior with their soccer players with disabilities (AA. M = 45.56) where feelings and personal rights are shared and clearly respected in a highly significant way (HA. M = 32.69). Furthermore, these coaches express a high capacity to put themselves in the place of their soccer players (AP. M = 33.23), recognize and understand emotional states, intentions, and perceptions expressed (CE. M = 33.55), allowing this situation, in a special way, to share with guarantees positive emotions (AE. M = 35.55) without the negative ones (EE. M = 23.80) affecting the actions that are generated within the coaching.

Concerning the type of coach profiles obtained, it is observed that the mean scores were significantly high, being above the average value, even some of these profiles, such as the “dialogic” (M = 26.92), “critical” (M = 32.10), “innovative” (M = 32.05) and “collaborative” (M = 33.95), reach scores close to the maximum range. This situation means that our coaches, from their coaching and professional experience, have achieved proactive professional development, establishing communicative and reflective flows with their soccer players (Table 1).

### 3.3. Level of Cognitive Empathy and Assertiveness according to the Profile of the Grassroots-Soccer Coach Working with Soccer Players Whose Express Some Type of Disability

Correlational analysis between the variables of empathy, assertiveness, and coach profile (Table 2) showed, on the one hand, that the assertiveness and empathy variables were linearly independent. An analysis with scatter plots did not show any other relationship between them. On the other hand, a low average positive correlation between the factors of the affective dimension of empathy and some coach models exists. Thus, the ES was positively related to the dialogic, innovative, and traditional profile; and the AE to the critical, innovative, and technological profile. In turn, the profiles of each factor were positively interrelated with each other.

To find out whether there were significant differences in the level of each of the coach profiles depending on assertiveness and empathy, this study did a factor analysis (ANOVA), taking the profiles as the dependent variable and the levels of the assertiveness and empathy factors (independent variables). ANOVA results showed significant differences in “empathic joy” for the critical profile F(2, 35) = 6.014, *p* = 0.006, η^2^ = 0.25 (0.411 **), innovative F(2, 35) = 6.739, *p* = 0.003, η^2^ = 0.27 (0.447 **) and technological F(2, 35) = 6.370, *p* = 0.004, η^2^ = 0.26 (0.477 **); as well as, in empathic understanding for the collaborative profile F(2, 35) = 6.022, *p* = 0.005, η^2^ = 0.24 (0.414 **). In all cases, the effect size was highly significant.

Then this study performed analyses using the Bonferroni test. This showed that participants with very high scores on empathic joy (M = 34.55, S.D. = 3.29) scored higher on the critical profile than those with medium scores (M = 29.22, S.D. = 3.07). Also, in the innovative profile, the coaches scored higher (M = 35, S.D. = 2.75) than coaches with medium (M = 29.33, S.D. = 2.69) and high (M = 31.56, S.D. = 4.19) scores. The same results were shown for the technological profile, where the coaches had very high scores (M = 32, S.D. = 4.79). Only in the collaborative profile is a highly significant score observed for empathic understanding (M = 33, S.D. = 3.44).

## 4. Discussion

The main purpose of our study is to identify and analyze the coaching profile of Laliga Santander Genuine grassroots-soccer coaches (learning and coaching context) who coach soccer players with disabilities in Spain. In this sense, the knowledge of the values and attitudes of these coaches coincides with Maestre, Garcés de los Fayos, Ortín, and Hidalgo [17] when the coaches manifest skills that allow them to express their emotions appropriately, without hostility or aggressiveness, with their group of soccer players in a reciprocal way (self- and hetero-assertiveness). This, in the words of Flynn, Hastings, Gillespie, McNamara, and Randell [11], expresses that the coaches must have a high level of respect and consideration for their own and others’ feelings, ideas, and behaviors within the coaching context.

In this way, these coaches have the facility to adopt an empathetic and understanding perspective based on the socioemotional states and situations that are generated in their soccer players [14]. This situation facilitates sharing concerns, motivations, and needs of positive emotional states through which beneficial situations are generated, ranging from the technical to the sporting part [5,10,15]. Acquiring skills and abilities will facilitate that soccer player learning can be transferred to other contexts, e.g., personal or labor [7,14,18,25].

Regarding the type of coach model analyzed, there is a very significant coincidence with other studies [8,9]. The situation is that these coaches are very organized and innovative regarding the distribution of technical and sporting work based on tasks and roles to be performed by the soccer players. Key references are creativity and improving coaching conditions with the group [5], allowing for greater performance and a higher degree of well-being [23]. Moreover, profiles such as the “critic” and “dialoger” are essential in the coaching program of the coaches because they show positive feedback among their soccer players based on instructions shared and agreed upon democratically, generating a climate of trust and satisfaction [7]. This affirmation coincides with Camacho et al. and Fraile Aranda et al. [7,8], who point out in their research that between 50% and 70% of the coach’s task with the athlete requires communication, which includes orientations for the performance of exercises, messages to maintain the efforts and motivation of the athlete, and the regulation or rectification of the motor execution.

Although the changing reality of sports practice demands that the grassroots-soccer coach must learn how to act in different situations, there are few studies on how coaches learn to reflect on their practical performance [9]. There are few studies on how coaches learn to reflect on their practical performance (critical profile) [17,19]. For example, to improve the ability to act critically in their practice, there are several action–research studies where a group of coaches ask themselves: What are the aims of my coaching program? What values do I defend with my education and teaching? What improvement can I include in my practice? What degree of interaction do I establish with the soccer players? From an education model based on a critical pedagogy, grassroots-soccer coaches learn to question everything that happens in their practice and adopt an attitude as learners of continuous improvement [19]. Coaches’ education should stimulate reflective and critical thinking that allows them to know themselves better and to seek coherence between their practice and their beliefs [1,5]. Likewise, to introduce among players a critical questioning of their practice, it is necessary to sensitize coaches to the search for solutions and the responsibility for their performance [2,6].

Another important aspect of this study is the level of cognitive empathy and assertiveness identified in the profiles of grassroots soccer coaches. We observe that profiles such as “dialogic”, “innovative”, and “traditional” can share the negative emotions that arise in the soccer players—connecting emotionally—[14,19] and being flexible to adapt to emotional changes from the acceptance of criticism, having as a key instrument, the communicative process generated for this purpose [20]. Furthermore, profiles such as “critical”, “innovative”, and “technological” express a climate of (self-)reflection seeking the effectiveness of the work through scientific knowledge shared in a generous and understanding way. This situation develops a comprehensive model for learning individual and collective technical and tactical behavior in grassroots soccer [15]. This activity, of a cognitive nature, on the one hand, represents one of the keys to active learning as it helps to act in a more active and creative way in their daily practice and, on the other hand, allows the soccer players to feel comfortable [1,14,15,17]. “Collaborative” coaches can “name” the emotions, connecting and recognizing the existing relationship with the experience of the soccer players to give meaning and significance to the coaching process [10]. Therefore, it is important to include in the initial education programs that coaches are taught to reflect on their practice collaboratively, generating shared knowledge. In this way, increased communication will generate a better body of knowledge [13]. Likewise, later on, in ongoing training, coaches should get used to sharing with other colleagues their didactic proposals, experiences, problems, possible solutions, etc., in collaborative communities [8]. Also, at technical and emotional levels, collaborative coaches can organize the work to be carried out based on a distribution of tasks and roles [9], effectively adapting to the needs and interests of the soccer players [15].

In general, each coach displays all the profiles more or less. This situation coincides with Fraile Aranda et al. [8] when this study says that “(…) although, in theory, it is relatively simple to define a coach, in practice, the differences are not easy to analyze because coaches work holistically. For this reason, it is difficult to differentiate between coaches” (p. 283).

Finally, there are some limitations to this study. Although the sample comprises coaching staff members of the professional soccer clubs of LaLiga Santander Genuine (Spain), it is not representative of the population, so the results are not generalizable. In this sense, this situation is key in this study because the assertiveness and empathy variables are linearly independent. The scatterplot analysis showed no other relationship, perhaps due to the small sample size. Furthermore, the questionnaire used to collect the data is one of the most widely used in research; it does not control for the social desirability that may occur, given that, for coaches, being empathetic is part of their professional identity [33]. Likewise, the study does not consider situational variables, for example, the characteristics of the person who empathizes with the grassroots soccer coach [28].

## 5. Conclusions

The different coaches analyzed from LaLigaSantander Genuine (Spain) during the 21/22 season have different profiles that are significantly related to empathic and assertive behaviors. At a descriptive level, this study concludes that the average age of the grassroots-soccer coaches is 34 years (range = 22–57). There is a majority of men (75%) who work professionally as grassroots-soccer coaches, compared to 25% female coaches. The surprising is that the majority of grassroots-soccer groups, with a disability, among their members are coached by women (75%) and 25% by men. Another important fact is that almost half of the coaches state that they have not received specific coaching to work with soccer players with disabilities.

From a more specific analysis, we can conclude that an innovative and dialogic profile is the “standard coach”. Other profiles like “technological” have less impact when working with disabled soccer players. This statement indicates that innovative and dialogic profiles of coaches are important to develop a style of communication in which the coach expresses, in a direct way, his ideas, feelings, and needs with his soccer players generating confidence, calm, and being honestly empathetic and respectful.

## Figures and Tables

**Table 1 sports-11-00161-t001:** Descriptive statistics for Assertiveness, Empathy, and Coaching Model measures.

	Factor	Media	Medium	D.S	Observed Range (Potential)
Assertiveness	Self-Assertiveness	45.56	47	6.632	30–59 (0–60)
	Hetero-Assertiveness	32.69	32	5.302	21–44 (0–45)
	Total	78.25	78	10.66	52–101 (0–105)
Empathy	Perspective Adoption	33.23	33	4.092	24–40 (8–40)
	Emotional Understanding	33.55	34	3.935	20–40 (9–45)
	Empathic Stress	23.80	23	6.378	10–35 (8–40)
	Empathic Joy	35.55	36	2.952	29–40 (8–40)
	Total	126.12	126	11.081	100–148 (33–165)
Model of coach	Dialoger	26.92	28	2.548	20–30 (6–30)
	Critic	32.10	32	3.837	24–40 (8–40)
	Innovative	32.05	33	3.973	22–40 (8–40)
	Collaborative	33.95	34	3.441	24–40 (8–40)
	Traditional	30.49	31	5.241	16–42 (9–45)
	Technological	27.05	29	6.236	10–40 (8–40)

**Table 2 sports-11-00161-t002:** Correlations between empathy, assertiveness, and coach profile variables.

	AA	HA	D	C	I	CLL	T	TCH
Dialoger	0.116	−0.046						
Critic	0.076	0.066	0.609					
Innovative	0.104	0.007	0.598	0.800 **				
Collaborative	0.247	0.291	0.411	0.554 **	0.424 **			
Traditional	0.196	−0.117	0.628	0.530 **	0.574 **	0.277		
Technological	0.058	−0.168	0.288	0.437 **	0.645 **	0.085	0.507	
PA	−0.144	−0.062	−0.051	0.124	0.097	0.076	−0.182	0.180
EQ	0.096	−0.058	0.094	0.178	0.236	0.414 **	0.061	0.075
ES	−0.223	−0.196	0.400	0.303	0.345 *	0.103	0.362 *	0.231
AE	−0.131	−0.006	0.265	0.412 **	0.447 **	0.159	0.114	0.477 **

Note 1: ** *p* < 0.01 * *p* < 0.05. Note 2: Abbreviations: AA (self-assertiveness); HA (hetero-assertiveness); D (dialoger); C (critic); I (innovative); CLL (Collaborative); T (traditional); TCH (technological); PA (perspective adoption); EQ (emotional understanding); ES (emphatic stress); AE (emphatic joy).

## Data Availability

Not applicable.

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
