# Peer review of "Soccer and Disability, Is It Possible? Analysis of the Learning and Coaching Context in Spain"

_sports, 2023, doi:10.3390/sports11090161_

Round 1

Reviewer 1 Report

This is an interesting study in coaches of soccer players with disabilities. Coaching athletes with disabilities is really different compared to coaching healthy athletes. Thus, i comment the Author for the effort to bring this matter on the surface of research. However, there are some major issues with the manuscript.

Major concerns:

1. There should be a balance between the extend of introduction and discussion. The majority of references are presented in the intro but the discussion lacks of in depth presentation and comparison of the results.

2. A separate paragraph of limitations is missing. Author placed the limitations in the conclusion paragraph. I suggest adding all the limitations in a last paragraph of discussion.

3. Since the majority of coaches were not working with disabled players (only 10%) how safe and reliable are the data in the current study?

Abstract: I suggest to Author to re-write the abstract including the main purpose, the methods, the main findings and a short take home message. Especially the conclusions needs improvement.

Line 8: Define "profile or coaching program". Is this refers to directions for these coaches?

Intro: The introduction is good, although the research questions are missing. There is a great load of details but the readers will lost. I am still not sure that this analysis is necessary, maybe focusing on the aim of the study.

Also, the intro reaches almost 3 pages. This might be tiring for readers.

Methods: Methods are good. Why Author included coaches from all sport clubs and not only coaches who work with disabled players? Pages 181-182 are in contrast to the participants of the study.

Was there any personal communication with participants? How they could ask questions and receive clarifications?

Results: Are the lines 216-218 refers to the 4 coaches only?

I suggest to Author not to use abbreviations inside Table 1. Instead, in Table 2 there should be an explanation of all abbreviations at the bottom of the table.

Discussion: Discussion needs an in depth analysis of the problem. It is too short compared to intro, while references from other sports might be helpful here.

Conclusions: This part is good but keep the first paragraph. Transfer the second one to the end of discussion and add limitations.

There are several points in the text where Authors are making mistakes in syntax and tenses. I kindly suggest to Authors an extensive editing in the whole manuscript.

Author Response

Dear Reviewer 1, please see the attachment. Kind regards

Reviewer 2 Report

1.      Please insert more elements from the results section into the abstract part.

2.      The proposal is that the paper's title remains as follows: Soccer and Disability: is it possible?: Analysis of the learning and coaching context in Spain.

3.      Lines 63-64. Camacho, Castejón-Riber, Requena, Camacho, Escribano, Gallego, Espejo, De Miguel-Rubio and Agüera [7]. Please leave just Camacho et al.

4.      The type and quality of the values that are acquired will depend on the agents of 82 socialisation in the sporting environment in which they develop, with the figure of the 83 coach having the greatest impact, playing a key role in the development and coaching of 84 individuals and athletes [1,14,15]. On a more careful search, two other sources were found that can support the background material:

·        Oh Y. The Relationship between Exercise Re-Participation Intention Based on the Sports-Socialization Process: YouTube Sports Content Intervention. Behav Sci (Basel). 2023;13(2):187. Published 2023 Feb 18. doi:10.3390/bs13020187

·        Sopa, I. S., & Szabo, D. A. (2015). Study regarding the importance of developing group cohesion in a volleyball team. Procedia-Social and Behavioral Sciences, 180, 1343–1350. https://www.researchgate.net/deref/https%3A%2F%2Fdoi.org%2F10.1016%2Fj.sbspro.2015.02.275

5.      At the end of the introduction, please  incorporate the novel elements of this research.

6.      Line 188. Please insert: SPSS statistical programme (IBM Corp. Released 2021. IBM SPSS Statistics for Windows, Version 28.0. Armonk, NY: IBM Corp).

7.      Lines 323-324. Finally, in general, each coach displays all the profiles more or less. This situation coincides with Fraile Aranda, Diego Vallejo and Boada i Grau [8]. Please leave just Fraile Aranda et al.

8.      Lines 343-344. Finally, the limitation of the sample. Please expand further on this subject in point 5. Limitations of the study

The English language requires moderate proofreading.

Author Response

Dear Reviewer 2, please see the attachment. Kind regards

Round 2

Reviewer 1 Report

Line 39: Change to made by previous studies [8,9].

No more comments.

Author Response

(The authors gave the same response as above.)

Reviewer 2 Report

Congratulations on your hard work in producing this scientific material!

Author Response

(The authors gave the same response as above.)
